# Relationship between Spinal Range of Motion and Functional Tests in University Students: The Role of Demographic Factors

**DOI:** 10.3390/healthcare12101029

**Published:** 2024-05-16

**Authors:** Nela Tatiana Balint, Bogdan Alexandru Antohe, Huseyin Sahin Uysal, Alina Mihaela Cristuță, Marinela Rață

**Affiliations:** 1Faculty of Movement, Sports and Health Science, “Vasile Alecsandri” University of Bacău, 600115 Bacău, Romania; tbalint@ub.ro (N.T.B.); rata.marinela@ub.ro (M.R.); 2Faculty of Sport Science, Burdur Mehmet Akif Ersoy University, 15500 Burdur, Turkey; hsuysal@mehmetakif.edu.tr

**Keywords:** spine, posture, students, deviations

## Abstract

Spinal disorders are some of the most prevalent health concerns, especially among students. Based on student demographics, this cross-sectional study evaluated the correlation between functional tests (FTs) and spinal range of motion (ROM). This study included 206 students (age = 19.85 ± 1.80 years) from the Vasile Alecsandri University of Bacău. Participants’ assessments were conducted using the following tests: (i) Ott, (ii) Schober, (iii) Stibor, (iv) finger-to-floor distance, (v) lateral flexion of the cervical and lumbar spine, and (vi) flexion of the cervical spine. Correlation analyses were evaluated using the Spearman correlation coefficient analysis. The results indicated a very strong relationship between lateral flexion of the lumbar spine on the left (LFLSL) and right (LFLSR) for all departments (*r* = 0.85 to 0.97, *p* < 0.05). There was a stronger relationship between FT results and spinal ROM for physical-education-department students compared to students from other departments (n = 17, *r* = −0.38 to 0.93, *p* < 0.05). There was no statistically significant correlation between FTs and spinal ROM based on age (*p* > 0.05). The study results provide evidence of the primary risk factors that predispose students to postural deviations. Practitioners and physiotherapists can utilize these values as a reference for potential therapeutic interventions.

## 1. Introduction

The posture defines the alignment or orientation of body segments while maintaining an upright position. On the other hand, incorrect posture refers to an irregular state in which the body does not support a steady position [1]. The World Health Organization (WHO) reports that spinal disorders and posture deficits are modern society’s most common health issues. Nowadays, due to sedentary lifestyles, spinal disorders are becoming increasingly common in people [2,3].

While many populations suffer from incorrect posture due to various factors, students are one of the groups affected [4]. Because the students are spending many hours in the same position while studying using a computer or cell phone, they are at high risk of posture deficits, spinal diseases, and having a limited range of motion [5]. Horodetska and Kuts reported that only 19.4% of boys and 12.3% of girls had a good study posture [6]. Although no evidence suggests that correct posture directly affects students’ academic performance, studies have found that physical education and movement-based study programs can positively impact spinal mobility and ROM [7,8]. One study revealed that only 31% of students were highly aware of their stance [9]. However, researchers observed that students with a positive attitude towards good posture inconsistently practice it [1]. Additionally, researchers have suggested that motivation levels may be related to good posture [10]. A systematic review reported a positive impact of school furniture dimensions on students’ performance and physical responses [11].

On the other hand, researchers observed that postural disorders significantly differed in children living in urban and suburban environments [12,13]. Similar studies have reported conflicting evidence regarding urban–rural differences in terms of the influence of living area on spinal ROM [14,15,16]. Finally, age and sex may also impact students’ posture [17]. Studies evaluating the relationship between spinal range of motion and sex have reported conflicting results. These contradictory results may be associated with variations in anatomical structures and the frequency of joint use in habitual physical activities between sexes [18].

Although many factors influence students’ posture, there has been insufficient empirical evidence for these factors in previous studies. Considering that the young human population also suffers from postural disorders, it is critical to assess students’ spinal ROM and uncover associated factors. Functional tests (FTs) such as the Schober test, Ott test, Stibor sign, and finger-to-floor test are commonly used to assess the range of motion, mobility, and overall functional capabilities of the spine [19]. Although these tests are traditionally used to determine the general population’s spinal ROM and public health, new evidence suggests that age does not affect test performances [20]. Therefore, these tests can also evaluate students’ spinal ROM.

This study aimed to evaluate the relationship between students’ spinal ROM and FTs and to determine the effect of demographic factors (students’ age, gender, living environment, university program) on this relationship. This study hypothesized that there would be a statistically significant relationship between students’ FTs and spinal ROM at different levels based on four variables (age, gender, living environment, and university program).

## 2. Materials and Methods

### 2.1. Study Design

The study was conducted using a single-blind, cross-sectional study design. The Quality Output Checklist and Content Assessment (QuOCCA) checklist was used to enhance the methodological quality of the study, and it is presented in Appendix A [21]. Additionally, the study protocol was pre-registered on the Open Science Framework (DOI: https://doi.org/10.17605/OSF.IO/9GC4B (accessed on 10 February 2024)), and details of the study files are provided on the website. The study was approved by the Vasile Alecsandri University Ethics Committee (Approval No. 7/1/22.02.2024) and was conducted in accordance with the Declaration of Helsinki. Informed consent was obtained from all participants involved in the study.

### 2.2. Participants

This study involved 206 students (106 males and 100 females, age: 19.85 ± 1.80 years, height: 166.86 ± 30.24 cm, and weight: 65.01 ± 14.11 kg) from the Vasile Alecsandri University of Bacău. These students were part of project CNFIS-FDI-2022-0087, which aimed to build a healthy student lifestyle and improve the quality of learning. Permission was obtained from the project manager to include the students in the study. The inclusion criteria for the study were as follows: (i) being a student at the Vasile Alecsandri University of Bacău, (ii) being 18 years or older, (iii) having a minimum class attendance of 80%, and (iv) having no physical limitations that could affect the results of the FTs. The exclusion criteria included severe scoliosis, disk hernia, Scheuermann kyphosis, lower limb injury, or height discrepancy and all the diseases that could interfere with the performance of the FTs.

This study assumed that there would be a small correlation between the students’ spinal joint range of motion and FTs. The minimum sample size of 202 participants was determined based on an a priori analysis using G*Power software (version 3.1, from the University of Dusseldorf, Germany). The analysis was conducted with the following parameters (correlation: bivariate normal model test, two-tailed, α = 0.05, β = 0.95, and *r* = 0.25). Details of the participant characteristics included in the study are presented in Table 1.

### 2.3. Procedures

The study was conducted at the Vasile Alecsandri University of Bacău in the Physical Therapy and Occupational Therapy research laboratory. Due to the high number of subjects, assessments were performed by five Physical Therapy MSc students, supervised by two physical therapists (T.B. and A.M.C.). The study collected data from each participant in a single session. However, the researchers conducted multiple measurement sessions due to the large number of subjects involved. All measurements were taken during the morning, specifically from 9 a.m. to 11 a.m. The room temperature was set to 22 degrees Celsius. Male participants were required to wear shorts for the test, while female participants were asked to wear shorts and a bra. To prevent interference with the results, all participants were instructed to refrain from eating or drinking anything for 60 min before the test and to avoid vigorous physical activity. The study protocol was explained to the participants before the measurements to prevent bias. The subjects were instructed to remain relaxed and cooperate with the physical therapist. All measurements were taken using metric tape, and the results were reported in centimeters (cm). The participants followed a standardized warm-up protocol, which included walking on a treadmill for 7 min, a 1 min cobra–cow stretch, and 1 min of child’s pose. The main goal of the warm-up was to eliminate any restrictions in the spinal range of motion, which could have been caused by insufficient synovial lubrication, before proceeding with the measurements.

### 2.4. Functional Tests

Cervical spinal flexion (CSF). Cervical spinal ROM was tested by measuring the distance from the suprasternal notch to the mandibular symphysis. The subject was instructed to bend their head forward while the physical therapist measured the distance between the specified points. If a subject can touch their chin to their chest, it indicates a full flexion ROM of the cervical spine [22,23]. Researchers reported that this measurement has an inter-rater reliability coefficient ranging from 0.92 to 0.88 [24].

Ott Test. The Ott test was used to measure the ROM of the thoracic spine in the sagittal plane. To perform the test, the seventh cervical vertebra was marked as C7, and a second point was placed on the thoracic spine, 30 cm below C7 (C7—30.0 cm↓). The distance between the points mentioned above was determined in the body’s upright position and during the torso’s maximal forward bend. A result of 33 cm or above (normal range of 3.0 cm) is considered significant [25,26,27].

Schober test. The Schober test was used to determine the ROM of the lumbar spine in the sagittal plane. The first step is to draw two horizontal lines at the L5 spinous process and the other 10 cm above it. The second step measures the distance between these points during maximal forward flexion. A less than 5 cm increase in length indicates a limited range of motion in the lumbar spine [28]. The test reliability is excellent, with an intraclass correlation of 0.96 and an interclass correlation of 0.90 [29].

Stibor sign. The Stibor sign examines the ROM of the lumbar and thoracic spine during maximal forward flexion. The Stibor sign is a cumulative measure of the other two tests, Schober and Ott. To perform the test, the evaluator measures the increase in the distance between two skin marks, one over the first sacral spinous process and the other over the C7 spinous process, after maximal forward bending [28,30].

Finger-to-Floor Distance (FFD). The subject stands with their feet about 15 cm apart. The distance between the fingertips and the floor is measured during maximal flexion of the spine and pelvis with the knees unbent. A higher value indicates greater trunk and lower limb muscle shortening, primarily affecting the hamstring muscles [15,31]. The test reliability is excellent, with an intraclass correlation of 0.999 [32].

#### Frontal Plane Tests

Lateral flexion of the lumbar spine. This was measured with the subject standing in a neutral position, with their feet 30 cm apart and their open hands against their thighs. The third finger was kept along the lateral side of the leg and marked with a pen on the thigh bilaterally before and after the measurements. Maximum active lateral flexion of the lumbar spine was measured once in each direction (left and right). A metric tape measured the distance in centimeters between the two marks [33]. Researchers reported an interobserver reproducibility of 0.74 and interobserver reproducibility of 0.96 for this test. Another study found an ICC value ranging from 0.920 to 0.983 [32,34].

Lateral flexion of the cervical spine. This was measured with tape from the tragus of the ear to the tip of the shoulder (acromion process) on the same side as the direction of neck movement. Compared with radiographs [35], the Spearman coefficient was 0.58 (*r* = 0.58). The ICC interrater reliability was 0.56 (0.31 to 0.74) and 0.44 (0.16 to 0.66) for interrater reliability [22,23].

### 2.5. Statistical Analysis

This study investigated the correlation between participants’ FT results and spinal ROM. Also, the potential factors that could influence the test outcomes were considered. To avoid assessment bias, the statistical analysis researchers were blinded to the data collection process. According to previous studies, it was assumed that four independent variables (university department, region of residence, sex, and age) could affect the results of FTs [36,37]. Six categorical variables were created based on university majors: (i) IT, (ii) Literature, (iii) Management, (iv) Engineering, (v) Physical Therapy, and (vi) Physical Education. The participants’ places of residence were categorized as either rural or urban. Finally, the age variable was analyzed as a numeric variable. The study data were presented with a correlation matrix, which included the correlation value (*r*) and significance value (*p*). The normality assumption of the data was checked using the Kolmogorov–Smirnov analysis, and it was observed that the data did not meet the normality assumption.

The relationship between FTs and independent variables was evaluated using the Spearman correlation coefficient analysis, and 368 analyses were conducted to report the results. The correlation coefficient was interpreted according to the following reference values: insignificant (<0.10), small (0.10 to 0.29), moderate (0.30 to 0.49), strong (0.50 to 0.69), very strong (0.70 to 0.89), or excellent (>0.90). Statistical analyses were conducted using R software (R Core Team, version 4.2.2, Vienna, Austria, https://posit.co/, accessed on 13 May 2024). The {ggplot2}, {patchwork}, and {metan} packages were selected for analysis and data visualization. All analyses were calculated with a 95% confidence interval, and the statistical significance level was set at α < 0.05. Analysis files and R codes used for this study are presented via OSF (https://doi.org/10.17605/OSF.IO/9GC4B (accessed on 10 February 2024)).

## 3. Results

The 206 participants who agreed to participate in this study demonstrated 100% adherence to the study protocol. There were no adverse events or injuries that occurred due to the study protocol, and no participants withdrew from the study.

### 3.1. Analysis of the Data from the Study Program and Functional Tests

The correlation between lateral flexion of the lumbar spine on the right (LFLSR) and lateral flexion of the lumbar spine on the left (LFLSL) scores was consistently positive and excellent across all groups (IT: r = 0.95, *p* = 0.001; Literature: r = 0.92, *p* = 0.001; Management: r = 0.97, *p* = 0.001; Engineering: r = 0.92, *p* = 0.001; Physical Therapy: r = 0.92, *p* = 0.001; Physical Education: r = 0.93, *p* = 0.001). Similarly, a positive correlation was found between the lateral flexion of the cervical spine on the left (LFCSL) and lateral flexion of the cervical spine on the right (LFCLR) scores of the five groups, ranging from very strong to excellent (IT: r = 0.88, *p* = 0.001; Literature: r = 0.77, *p* = 0.001; Management: r = 0.91, *p* = 0.001; Engineering: r = 0.88, *p* = 0.001; Physical Education: r = 0.84, *p* = 0.001). However, there was a moderate correlation between the LFCSR and LFCSL scores of the Physical Therapy group (r = 0.43, *p* = 0.01). 

On the other hand, it was revealed that the correlation between the FT results of the groups varied. While the correlation between the eight FTs in the IT group was moderate (LFLSL and LFCSR: r = 0.44, *p* < 0.05; LFLSR and LFCSR: r = 0.40, *p* < 0.05; LFLSR and LFCSL: r = 0.45, *p* = 0.01; LFCSL and CSF: r = 0.46, *p* = 0.01), the correlation between the four FTs was strong (LFLSL and LFCSL: r = 0.53, *p* = 0.01; LFCSR and CSF: r = 0.51, *p* = 0.01). A strong correlation was found among the six FTs in the Literature group (LFCSL and FFD: r = 0.52, *p* < 0.05; LFCSR and FFD: r = 0.52, *p* < 0.05; Stibor sign and Schober index: r = 0.65, *p* = 0.01). Additionally, the Ott sign and four FTs demonstrated a statistically significant, strong negative correlation (Schober index: r = −0.51, *p* < 0.05; Stibor sign: r = −0.56, *p* < 0.05; LFCSL: r = −0.60, *p* < 0.05; LFCSR: r = −0.51, *p* < 0.05). In the Management group, there was a very strong to excellent positive correlation between lateral flexion of the lumbar spine and lateral flexion of the cervical spine (r = 0.86 to 0.97, *p* = 0.001). While the Schober index revealed a strong negative correlation with two FTs (LFCSR: r = −0.56, *p* < 0.05; LFLSR: r = −0.53, *p* < 0.05), the Stibor sign exhibited a strong positive correlation with two FTs (CSF: r = 0.58, *p* < 0.05; Schober index: r = 0.69, *p* = 0.01). While the correlation between the Stibor sign and Schober index was strong for the Engineering group, no statistically significant results emerged among other FTs. The Physical Therapy group indicated a moderate negative correlation between the Schober index, Stibor index, Ott sign, and LFCSL (r = −0.39 to −0.42, *p* = 0.01). On the other hand, there was a strong relationship between the Stibor index and the Schober index (r = 0.68, *p* = 0.001). Additionally, small-to-moderate statistically significant results were found in the 16 FTs. Finally, in the Physical Education group, there was a strong relationship between the Stibor and Schober indices (r = 0.64, *p* = 0.001). A negative correlation was found between the Schober index and lateral flexion of the cervical spine (r = −0.38, *p* = 0.01). Additionally, 24 relationship analysis results revealed statistically significant findings. 

All the other correlation analyses conducted did not reveal any statistically significant results. The analysis results relating to the study program and FTs are presented in Figure 1.

### 3.2. Analysis of the Data from the Region of Residence and Functional Tests

While there was an excellent positive correlation between LFLSR and LFLSL scores of rural-region participants (r = 0.91, *p* = 0.001), there was also a very strong positive correlation between LFCSL and LFCSR (r = 0.86, *p* = 0.001). Similarly, there were positive moderate correlations between the flexion cervical spine and the flexion lumbar spine tests (r = 0.43 to 0.53, *p* = 0.001). On the other hand, a strong positive correlation was found between the Schober index and the Stibor sign (r = 0.62, *p* = 0.001). Finally, ten small levels of significant correlations were noted for the rural group. 

Similar results were observed in participants residing in urban areas. While an excellent positive correlation was found between LFLSR and LFLSL (r = 0.94, *p* = 0.001), a very strong positive correlation was also found between LFCSR and LFCSL (r = 0.79, *p* = 0.001). Additionally, the results revealed a strong positive correlation between the Stibor sign and the Schober index (r = 0.66, *p* = 0.001). A moderate positive correlation was observed between the CSF, LFCSL, LFCSR, LFLSL, and LFLSR tests in eight analyses (r = 0.29 to 0.41, *p* = 0.01). Although significant differences were found in six FT analyses, the level of correlation was determined to be small. The analysis details for the region of residence are presented in Figure 2.

### 3.3. Analysis of the Data between Sex and Functional Tests

The correlation between LFLSR and LFLSL was positive and excellent for both sexes (male: r = 0.92, *p* = 0.001; female: r = 0.91, *p* = 0.001). A very strong positive correlation between LFCSL and LFCSR was found in the analysis results (male: r = 0.79, *p* = 0.001; female: r = 0.74, *p* = 0.001). A strong positive correlation was also found between the Stibor sign and the Schober index (male: r = 0.56, *p* = 0.001; female: r = 0.70, *p* = 0.001). The results of 13 FT analyses of male participants indicated small but statistically significant correlation results, while 12 analyses of female participants also revealed small yet statistically significant results. Details of the relationship between FTs according to sex are presented in Figure 3.

### 3.4. Analysis of the Data between Age and Functional Tests

Although the age variable was analyzed with eight FTs as numerical variables, no statistically significant relationship was observed between age and any FT. Details about the relationship between age and FTs are presented in Figure 4.

## 4. Discussion

This study aimed to investigate the relationship between FT tests assessing spinal ROM and various demographic factors among students, including their university department, region of residence, sex, and age. The findings revealed that the study department was an important risk factor for the students, since the Physical Education Department had better results in the FT evaluation, compared with students from other departments. We found no difference in FT values and age of the subjects. Also, students who live in rural areas had better FT results than students who live in urban areas. All these results are explained in detail below.

Upon examining the correlation between the university department variable and FTs, it was discovered that students majoring in Literature or Management exhibited a stronger correlation with the lateral flexion of the lumbar and cervical spine tests. Consistent with previous studies, a high prevalence of postural deviations in the frontal plane of the spine was observed among students [4,38]. Furthermore, previous studies have indicated a higher incidence of scoliosis among female students [39,40]. 

Literature, Management, and Engineering students are more likely exhibit correlations with FTs on the sagittal plane. There was a slight difference in the direction of students from urban areas in the sagittal plane. In addition, there were many positive correlations for the FTs in the female group, especially for the Schober test and the cervical spinal flexion test. Kyphosis is a common spinal deformity in students, especially females, according to the literature. Also, forward head posture was found in 63.96% of students [41,42,43]. These findings supported current study results since most students were between 18 and 21 years old, and 67% were female. It has been observed that students majoring in Literature, Management, and Engineering are more likely to show a higher correlation with sagittal plane FTs. However, there was a slight difference in the results for students from urban areas in the sagittal plane. The characteristic of these departments is sitting posture, which moves forward the body’s center of gravity and kyphosis appears [44]. In addition, female groups indicated a positive correlation with FTs, especially for the Schober and cervical spinal flexion tests. The literature also suggests that spinal deformities such as kyphosis commonly affect students, particularly female group [45]. Furthermore, it was observed that 63.96% of students had a forward head posture [46], and 67% of them were female.

The risk factors associated with this type of deviation, which are characteristics of students, include poor sitting posture, muscle imbalances, sitting away from the table, and poorly positioned lower limbs [47,48]. Prolonged smartphone usage is another critical factor in the correlations found in the cervical spinal flexion test [49,50]. Excessive smartphone use can shift the head’s center of gravity forward. 

Frontal and sagittal plane dysfunctions of the spine, such as forward head posture, kyphosis, and scoliosis, can increase the risk of degenerative disk disease, disk hernia, vertebral body compression, and zygapophysial joint arthritis [43,51,52]. The study found no correlations between variables when analyzing the results based on age. This may be because the study’s subjects were all of a similar age, ranging from 19 to 23 years old. Although the literature suggests that age is a risk factor for spinal deformities, it usually refers to those aged between 13 and 15 years old or older adults [40,53,54]. Additionally, the absence of correlations does not necessarily indicate the lack of future spinal deformities. Our subjects are young people who spend more than two hours on their smartphones and computers and are vulnerable to health problems associated with prolonged device use [55]. Age can be an essential factor for posture deformities. Still, it should be considered along with other risk factors such as sex, environment, level of physical activity, etc.

The lack of significant correlations between age and FT outcomes suggests that age may not be this population’s primary determinant of spinal range of motion [20]. The study findings indicate that factors other than age may have a greater influence on spinal health among young individuals. The studies on spinal mobility in different sexes have produced mixed results. While some studies have found no significant differences in spinal mobility between sexes [14,15,16], other studies suggest that sex differences in spinal mobility may be influenced by factors such as job type, disease, and spinal level. The findings indicate significant differences in postural deviations between males and females, with a stronger correlation observed in females. The strength of muscles and fascia, as well as the stiffness of joints, are vital factors that affect men’s spinal health [56]. Moreover, anatomic variations in spine pedicles, which are more prominent in males, may explain these differences [56]. While physically demanding occupations can help maintain men’s spinal health, they also pose a risk for reduced intervertebral disc height and range of motion [57]. On the other hand, the primary risk factor for spinal deformities in females is their increased range of motion compared to males [58,59], as well as static jobs and hormonal laxity [60].

It is a commonly accepted fact that rural residents tend to engage in more physical activity than those who live in urban areas [61]. This indicates that the environment in which one lives can be a risk factor for spine-related health problems. According to the study results, urban dwellers are more likely to suffer from spine-related issues due to stress, pollution, and sedentary jobs. These conditions can significantly increase the chances of developing spinal diseases [38]. On the other hand, rural residents who tend to be more physically active need to be cautious not to indulge in excessive physical activity as this can also lead to spine-related problems. Overloading the spine with excessive physical activity can result in health problems, too. Furthermore, the positive correlations observed between specific FTs, such as lateral flexion of the lumbar spine and cervical spine, among different student groups suggest the presence of consistent patterns in spinal mobility among individuals with diverse living environments and lifestyles [62,63]. 

### Limitations

Since the subjects’ average age was 19.85 ± 1.80 years, we consider this a limitation of this study. It will be interesting to apply the same measurement protocol to a younger or elderly population. Also, the study subjects were students. We cannot state whether the results of this study can be applied to other population categories. This study did not measure the rotation and extension of the spine, which would have doubled the number of variables and made it challenging to include all the data in this study. The researchers should validate the tape measure against the gold radiography standard. However, the study measurements were validated through inter- and intraclass correlations, which support the accuracy of the study data. Furthermore, subjecting healthy volunteers to radiation raised ethical concerns. It is worth noting that the results of the study may not be directly applicable to young students. Although this study evaluates the relationship between spinal ROM and FTs, the existing relationship can be modeled using more data with quantile regression, complex network, or principal component analysis. With these analyses, the relationship between spinal ROM and FTs can be revealed more clearly.

## 5. Conclusions

It has been observed that postural asymmetries in the frontal plane of study participants can be influenced by various factors such as age, sex, environment, and living conditions. These findings emphasize the necessity of interventions to enhance postural habits and decrease the prevalence of postural changes in students.

Moving forward, longitudinal studies are needed to further elucidate the underlying mechanisms driving these correlations and to inform the development of effective preventive strategies tailored to individual needs and circumstances. Variations were observed in the strength and direction of correlations between different FTs within each student group. This variability may be attributed to many factors, including individual biomechanical differences, habitual movement patterns, and environmental influences.

## Figures and Tables

**Figure 1 healthcare-12-01029-f001:**
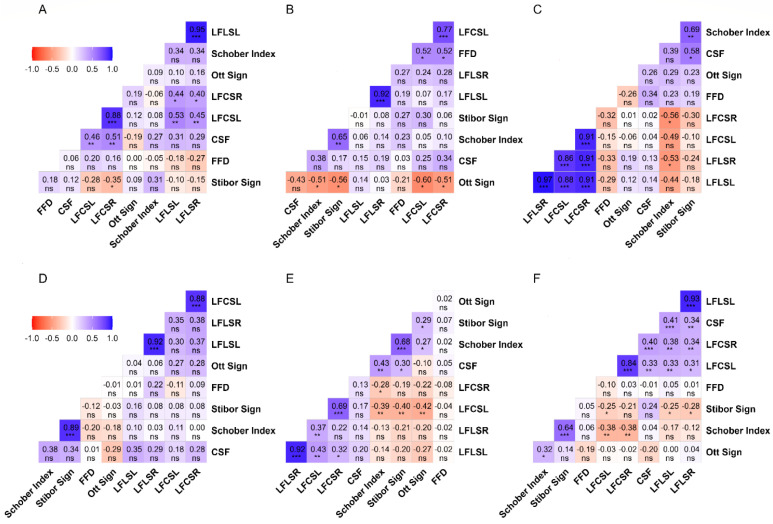
The results of the correlation between FTs and spinal ROM tests based on departments. Legend. (**A**): Information Technology students; (**B**): Literature students; (**C**): Management students; (**D**): Engineering students; (**E**): Physical Therapy students; (**F**): Physical Education students; FFD: Finger-to-floor distance; LFLSL: Lateral flexion lumbar spine left; LFLSR: Lateral flexion lumbar spine right; LFCSL: Lateral flexion cervical spine left; LFCSR: Lateral flexion cervical spine right; CSF: Cervical spinal flexion; ns = *p* ≥ 0.05; * = *p* < 0.05; ** = *p* < 0.01; *** = *p* < 0.001. The correlation matrix uses color gradients to represent the correlation between dependent variables. The negative correlation of red or near-red color tones, the positive correlation of blue or near-blue hues, and color tones close to white and white indicate a neutral correlation. While the correlation value is displayed in the boxes, the statistical significance of this value is indicated below the value.

**Figure 2 healthcare-12-01029-f002:**
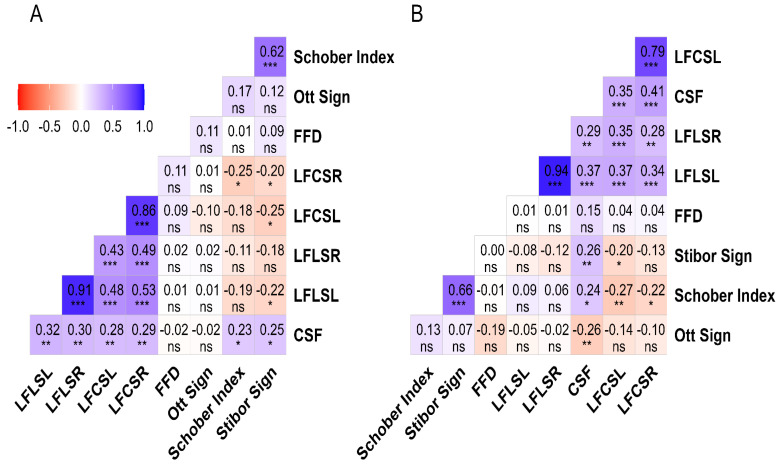
The results of the correlation between FTs and spinal ROM tests based on living environment. Legend. (**A**): Rural region; (**B**): Urban region; FFD: Finger-to-floor distance; LFLSL: Lateral flexion lumbar spine left; LFLSR: Lateral flexion lumbar spine right; LFCSL: Lateral flexion cervical spine left; LFCSR: Lateral flexion cervical spine right; CSF: Cervical spinal flexion; ns = *p* ≥ 0.05; * = *p* < 0.05; ** = *p* < 0.01; *** = *p* < 0.001. The correlation matrix uses color gradients to represent the correlation between dependent variables. The negative correlation of red or near-red color tones, the positive correlation of blue or near-blue hues, and color tones close to white and white indicate a neutral correlation. While the correlation value is displayed in the boxes, the statistical significance of this value is indicated below the value.

**Figure 3 healthcare-12-01029-f003:**
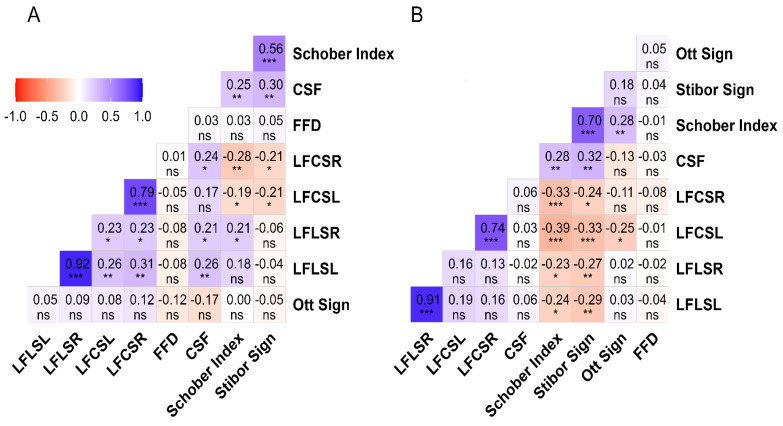
The results of the correlation between FTs and spinal ROM tests based on sex. Legend. (**A**): Male; (**B**): Female; FFD: Finger-to-floor distance; LFLSL: Lateral flexion lumbar spine left; LFLSR: Lateral flexion lumbar spine right; LFCSL: Lateral flexion cervical spine left; LFCSR: Lateral flexion cervical spine right; CSF: Cervical spinal flexion; ns = *p* ≥ 0.05; * = *p* < 0.05; ** = *p* < 0.01; *** = *p* < 0.001. The correlation matrix uses color gradients to represent the correlation between dependent variables. The negative correlation of red or near-red color tones, the positive correlation of blue or near-blue hues, and color tones close to white and white indicate a neutral correlation. While the correlation value is displayed in the boxes, the statistical significance of this value is indicated below the value.

**Figure 4 healthcare-12-01029-f004:**
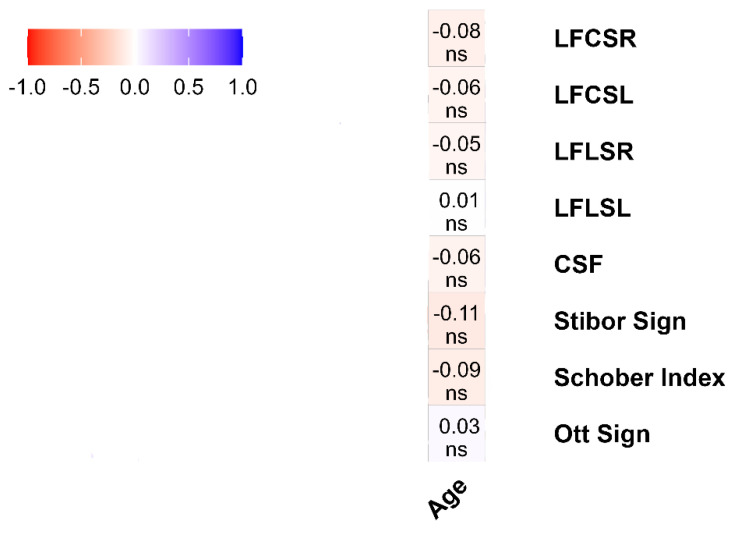
The results of the correlation between FTs and spinal ROM tests based on age. Legend. LFLSL: Lateral flexion lumbar spine left; LFLSR: Lateral flexion lumbar spine right; LFCSL: Lateral flexion cervical spine left; LFCSR: Lateral flexion cervical spine right; CSF: Cervical spinal flexion; ns = *p* ≥ 0.05. The correlation matrix uses color gradients to represent the correlation between dependent variables. The negative correlation of red or near-red color tones, the positive correlation of blue or near-blue hues, and color tones close to white and white indicate a neutral correlation. While the correlation value is displayed in the boxes, the statistical significance of this value is indicated below the value.

**Table 1 healthcare-12-01029-t001:** Characteristics of the participants.

Variable	Subgroups	n	%	Total Number of Participants
Sex	Male	106	51.45	206
	Female	100	48.55	
Living region	Urban	105	50.98	
	Rural	101	49.02	
Department	IT	33	16.03	
	Literature	16	7.67	
	PT	55	26.70	
	PE	64	31.10	
	Engineering	23	11.20	
	Management	15	7.30	

Note. IT: Information Technology; PT: Physical Therapy; PE: Physical Education.

## Data Availability

https://doi.org/10.17605/OSF.IO/9GC4B.

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
