# Peer review of "Relationship between Spinal Range of Motion and Functional Tests in University Students: The Role of Demographic Factors"

_healthcare, 2024, doi:10.3390/healthcare12101029_

Round 1

Reviewer 1 Report

Comments and Suggestions for Authors

This manuscript entitled “Relationship between spine range of motion and functional tests in university students: The role of demographic factors” was primarily aimed to assess the correlation between function tests and spinal range of motion among the university student population. The authors bring an interesting study, but there are still some problems that cannot up this article to a publishing level. Suggestions are listed in the specific comments below.

Specific comments:

1.     In the Abstract part, since authors investigated the correlation between FT and spinal ROM based on age, the age range of participants should be provided in the participant information of the abstract part. Otherwise, this conclusion by the authors would mislead the reader into thinking that there is no correlation between FT and spinal ROM in all age groups

2.     In the Introduction part, line 34-35, “Nowadays, due to our sedentary lifestyle, spinal disorders are becoming increasingly common in people.” It is recommended to cite the relevant papers to support this statement.

3.     In the Introduction part, line 40-41, “Therefore, various factors can influence students' posture.” Can you be more specific about what are the various factors? Based on your previous descriptions, it is still unclear about these factors.

4.     In the Results part, line 196, “The correlation between LFLSR and LFLSL scores was…” Abbreviations that exist in articles should be explained when they first appear, such as LFLSR and LFLSL. Please give official explanations of the abbreviations in the article. Please check this problem throughout the results part.

5.     In the Discussion part, it is recommended to provide a brief description of the main findings in the first paragraph of the discussion part. Some recently studies could be added in the discussion, such as:

Physiological Features of Musculoskeletal System Formation of Adolescents Under the Influence of Directed Physical Training’, Physical Activity and Health, 7(1), p. 1–12. Available at: https://doi.org/10.5334/paah.217.

6.     In the Discussion part, line 337-339, “It has been observed that students majoring in Literature, Management, and Engineering are more likely to show a higher correlation with sagittal plane functional tests. However, there was a slight difference in the results for students from urban areas in the sagittal plane.” Can you explain more the potential reasons for this difference.

7.     In the Discussion part, line 351-353, “According to researchers, prolonged use of smartphones is also a critical factor in the correlation with cervical spine flexion tests.” I think it is redundant, authors have already mentioned it in previous descriptions.

8.     In the Discussion part, line 401-402, “It is worth noting that the results of our study may not be directly applicable to young students.” Are the results also applicable for the old?

Comments on the Quality of English Language

Minor editing of English language required

Author Response

Dear reviewer,
Thank you for taking your time to evaluate our article.
Please see the answers to your comments in the attachment.

Reviewer 2 Report

Comments and Suggestions for Authors

I really appreciate the prepared topic. The introduction is sufficient and the section on methods and materials is described in great detail and exhaustively. Congratulations.
The results are presented in a clear and transparent manner.
The discussion is exhaustive and complete.
The conclusions lack research limitations that are obvious and result from the selection of the sample - students of one university.

Author Response

(The authors gave the same response as above.)

Reviewer 3 Report

Comments and Suggestions for Authors

Thank you for submitting send manuscript “Relationship between spine range of motion and functional tests in university students: The role of demographic factors”. The manuscript have got potential to be good scientific paper, but in this form need to be change. Presenting the research results of the submitted scientific article required many sacrifices and logistics and organization of scientific research. After reviewing mentioned above scientific paper I would like to require some correction and advices. Below I am sending You outline possible points for revision in the chronological order of the manuscript.

Introduction - please add citation/s to the sentences (line 35).

All scientific paper - in the whole scientific paper, I propose to standardize the citations. If authors record more than one citation please record them as follows [1,2] and not as is the case throughout the scientific article [1], [2].

Introduction - I propose to add to  change the following sentence by starting (line 39) not from “A study reported…” but for example “Horodetska study showed…”

All scientific paper - in the whole scientific paper, I propose not to use “we hypothesized… (line 69), or not to use the following words: we, you, but for example it was found, The study showed…”

Introduction - please add citation/s to the following sentence “Finally, age and sex may also impact students’ posture”.

Material and methods (section 2.1.) – do not use doi number but the citation of it (line 80).

Material and methods (section 2.3.) – it is not important in details what the participants were wearing (line 111-112).

Material and methods (section 2.3.) – what was the speed of treadmill for the participants (line 119-120).

Results (section 2.3.) – please correct the sentence (line 255-256). “…in the ten relationship…”

Discussion – line 332-333 The sentence start from “Moreover, female group…” not women.  

References – please check the following references: 6, 11, 25, 30, 41 – lack of pages of scientific journals.

Comments on the Quality of English Language

A scientific article written in English is of a good standard.

Author Response

(The authors gave the same response as above.)

Reviewer 4 Report

Comments and Suggestions for Authors

I congratulate the authors for developing the article: "Relationship between spine range of motion and functional 2 tests in university students: The role of demographic factors".

Follow my comments under:

1- Title section:

My personal opinion. What do the authors think about removing the subtitle from the title? My opinion is to delete the subtitle and simply reset the title. Please consider it, authors.

2- In the introduction section: Line 85

The authors presented information about the participants. However, the individual characteristics of the study subjects need to be presented in a table.

3- Methods section:

If an abbreviated term is reused in the title or abstract, it must be described as an abbreviation throughout the body. For example,

Line 180

"The relationship between functional tests and independent variables ~~"

In the case of the above sentence,

"The relationship between FT and independent variables ~~"

It may be modified as follows.

4- Many minor mistakes such as typos and paragraph division were detected throughout the paper. For example,

Lines 194-195, Line 403 (empty) etc

Once again, the authors need to revise them carefully.

5- The authors must re-examine the references.

Comments on the Quality of English Language

I congratulate the authors for developing the article: "Relationship between spine range of motion and functional 2 tests in university students: The role of demographic factors".

Follow my comments under:

1- Title section:

My personal opinion. What do the authors think about removing the subtitle from the title? My opinion is to delete the subtitle and simply reset the title. Please consider it, authors.

2- In the introduction section: Line 85

The authors presented information about the participants. However, the individual characteristics of the study subjects need to be presented in a table.

3- Methods section:

If an abbreviated term is reused in the title or abstract, it must be described as an abbreviation throughout the body. For example,

Line 180

"The relationship between functional tests and independent variables ~~"

In the case of the above sentence,

"The relationship between FT and independent variables ~~"

It may be modified as follows.

4- Many minor mistakes such as typos and paragraph division were detected throughout the paper. For example,

Lines 194-195, Line 403 (empty) etc

Once again, the authors need to revise them carefully.

5- The authors must re-examine the references.

Author Response

(The authors gave the same response as above.)

Round 2

Reviewer 1 Report

Comments and Suggestions for Authors

All my questions have been well addressed, now, I recommend to accept it. 

Comments on the Quality of English Language

Minor editing of English language required